# HINDSIGHT: POSTERIOR-GUIDED TRAINING OF RE-TRIEVERS FOR IMPROVED OPEN-ENDED GENERATION

**Ashwin Paranjape, Omar Khattab,**
**Christopher Potts, Matei Zaharia & Christopher D. Manning**
Stanford University
`{ashwinp,okhattab}@cs.stanford.edu`

## ABSTRACT

Many text generation systems benefit from retrieving passages from a textual knowledge corpus (e.g., Wikipedia) and using them to generate the output. For open-ended generation tasks, like generating informative utterances in conversations, many varied passages $z$ are relevant to the context $x$ but few are relevant to the observed next utterance $y$ (label). For such tasks, existing methods (that jointly train the retriever and generator) underperform: during training the top-k context-relevant retrieved passages might not contain the label-relevant passage and the generator may hence not learn a preference to ground its generated output in them. We propose using an additional guide-retriever that also conditions on the observed label $y$ and "in hindsight" retrieves label-relevant passages during training. We maximize the evidence lower bound (ELBo) to jointly train the guide-retriever $Q(z|x, y)$ with the standard retriever $P_\eta(z|x)$ and the generator $P_\theta(y|x, z)$ and find that ELBo has better inductive biases than prior work. For informative conversations from the Wizard of Wikipedia dataset, with our posterior-guided training, the retriever finds passages with higher relevance in the top-10 (23% relative improvement), the generator's responses are more grounded in the retrieved passage (19% relative improvement) and the end-to-end system produces better overall output (6.4% relative improvement).

## 1 INTRODUCTION

In knowledge-intensive NLP tasks, models must use open-domain knowledge to answer questions (Kwiatkowski et al., 2019; Joshi et al., 2017), fact-check claims (Thorne et al., 2018) or engage in informative conversations (Dinan et al., 2019; Zhou et al., 2018). State-of-the-art models for open-domain question answering are *retrieval-augmented*: they extract relevant passages from a human-readable corpus (e.g., Wikipedia) using a learned *retriever* and process it with a task-specific *reader*. If the relevant passage is known (e.g., human-annotated gold passage), the retriever can be supervised with it. In this work we consider *open-ended* generation tasks where the gold-passages are unknown. Figure 1 illustrates this **one-to-many** setting: for a conversational context $x$, many relevant passages $z$ (dubbed **context-relevant** passages) could have generated many coherent responses. But only $z_{gold}$ (dubbed **label-relevant** passage) generates the observed target output $y$. Had we known $z_{gold}$ corresponding to the target output, we could have supervised the retriever with $z_{gold}$ and trained the generator conditioned on $z_{gold}$ – but we don't!

Current methods for retrieval-augmented generation (Lewis et al., 2020) work well for short-answer QA-like tasks: Natural Questions (Kwiatkowski et al., 2019) or fact-checking (Thorne et al., 2018). Lewis et al. (2020) use the generator's probability distribution $P_\theta(y|x, z)$ as a proxy for label relevance and train the retriever $P_\eta(z|x)$ by marginalizing $p(y|x)$ over retrieved documents $z$: $P(y|x) = \sum_{z \in \text{top-k}(P_\eta(.|x))} P_\eta(z|x)P_\theta(y|x, z)$. However, for one-to-many tasks, this objective leads to suboptimal solutions: the generator is less grounded in the retrieved passages (Figure 3), the retriever performance saturates at low recall (Figure 3), and the top-k retrieved passages exclude many label-relevant passages weakening the supervision during training (Table 1).

In our work, as a proxy for $z_{gold}$, we train a separate **guide-retriever** model to find label-relevant passages. The **guide-retriever** uses both the input $x$ and the output $y$ and is represented by

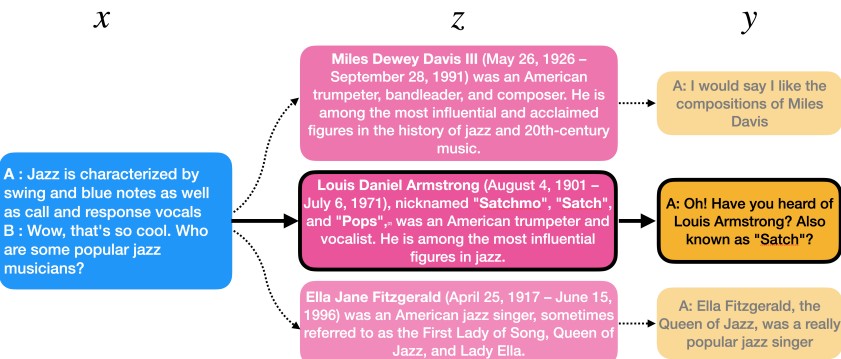

Figure 1: A conversational turn with many plausible responses. The input (blue) can be answered based on 3 equally *context-relevant* passages but only one possible response (yellow) is observed in the training set based on only one of the pink *label-relevant* passages (outlined in black).

the label-posterior distribution $Q(z|x,y)$ that captures label-relevance in "hindsight". Modeling the label-posterior distribution $Q(z|x,y)$ with a full-fledged retriever generalizes weak supervision approaches and retrieves label-relevant passages from the entire collection. We jointly optimize the retriever, posterior-guide, and generator using the evidence lower bound (ELBo): $\mathbb{E}_{z_i \sim Q(.|x,y)}[\log P_\theta(y|x,z)] - D_{\mathrm{KL}}(Q|P_\eta)$. While the objective function is a lower bound, it encodes biases that improve joint-training on open-ended tasks: (1) conditioning the generator on the passages weighted by their label-relevance (from the label-posterior distribution) increases grounding and (2) training the retriever with a mode-seeking reverse-KL divergence encourages it to match some modes with the guide (label-relevant passages), with a lesser penalty for matching other modes (other context-relevant passages).

Our main contribution is a complete HINDSIGHT training system that: (1) uses a guide-retriever to provide a stronger learning signal for both the generator and the retriever and (2) is amenable to index-updates with iterative closed-set training (Section 3). To evaluate one-to-many open-ended generation tasks, it is insufficient to just evaluate the end-to-end performance of the joint system. Thus, we also evaluate the individual models (retriever and generator) and at varying passage depths. Using HINDSIGHT on the Wizard of Wikipedia dataset of informative conversations: the retriever finds more relevant passages with a 23% relative improvement (r.i.) in success@10 (i.e., is the label-relevant passage among the top-10 retrieved passages?), the generator is more grounded with 19% r.i. in Novel-F1 overlap with the top-1 retrieved passage (i.e., its overlap with the retrieved passage excluding words that are common or in the input) and the combined system is overall better with a 6.4% r.i. in Novel-F1@1 overlap with the gold utterance (the best matching generation when considering top-1 retrieved passage). HINDSIGHT also improves performance on the MS-MARCO NLGen dataset, a one-to-one free-form QA task.

## 2 BACKGROUND

**Open-domain Question Answering** In the reading comprehension task, a passage is given and the models extract the answer span from it. In Open-domain QA (a.k.a. open-QA) no such passage is given; the models are expected to extract the answer from a large document corpus. Dr. QA (Chen et al., 2017), the first neural system for factoid open-QA, used an off-the-shelf retriever (e.g., TF-IDF, BM25) to find relevant passages and trained a reader to extract the answer span. Now, trainable neural retrievers have replaced the classical term-matching retrievers. Here, pre-trained models (like BERT) embed the document corpus and the query into a single vector space and efficient nearest-neighbour search algorithms (Jegou et al., 2010; Johnson et al., 2017) find the relevant passages corresponding to the query. The neural retriever can be trained variously: pretraining with the inverse cloze task then weakly supervising using span matches (Lee et al., 2019), using gold passages with in-batch negatives (Karpukhin et al., 2020), and retrieval-guided supervision with span-based positives (Khattab et al., 2021).

**Open-ended Generation**     Natural language generation tasks provide some input (sequence of tokens, image) and expect the system to produce another sequence of tokens (or word-pieces) as output. An open-ended task accepts a higher diversity of generations. Factoid question-answering with a single correct short answer is less open-ended than free-form long answers. Machine translation accepts a few correct translations (Bojar et al., 2014) but they are less diverse than informative dialogue, where the speakers can lead the conversation in many different directions (Dinan et al., 2019), making it more open-ended. Many more generation tasks such as summarization (Narayan et al., 2018) and story generation (Mostafazadeh et al., 2016) lie on this spectrum.

**Retrieval for Language Modeling**     Khandelwal et al. (2020) retrieve similar contexts from the training set at each time-step and increase the likelihood of tokens that were predicted in similar contexts. Guu et al. (2020) instead pre-train a retrieval-augmented masked language model using salient-span masking and fine-tune it on downstream QA tasks.

**Using labels for direct supervision**     Zheng et al. (2020) use term-overlap with the label as a heuristic to identify the gold-passage from a small passage set ($\sim 50$) and train a reranker. Prior work has also modeled the posterior of various probabilistic models (Lian et al., 2019; Kim et al., 2020; Zhan et al., 2021) or used reinforcement learning (Zhao et al., 2020) to improve knowledge selection from the small passage set. In Zheng et al. (2021), the authors increase grounding by using the label to reweigh passage tokens and in Cai et al. (2019) they increase grounding by feeding a corrupted version of the label to the generator as a stand-in for the label-relevant passage during training.

**Retrieval-Augmented Generation**     Lewis et al. (2020) introduce retrieval-augmented generation, where, for input $x$ and output $y$, a retriever finds top-k passages ($z$) from a corpus and jointly train it with a generator ($P_\theta$) by maximizing the likelihood of the output marginalized over the top-k documents. In this work, we refer to this loss function as the MARGINALIZEDLOSS:

$$P(y|x) = \sum_{z \in \text{top-k}(P_\eta(.|x))} P_\eta(z|x) P_\theta(y|x, z) \tag{1}$$

Here $P_\theta(y|x, z)$ is conceptually utilized in two roles: first, supervising the retriever (i.e., teaching the retriever to score label-relevant passages higher than other passages) and keeping the generator grounded (i.e., maximizing the probability of the target output given the context-relevant passages). In the next section we introduce a guide-retriever to capture the label-relevance and we train it using ELBOLOSS, a lower bound to MARGINALIZEDLOSS, that has better inductive biases.

## 3   TRAINING WITH HINDSIGHT

To identify label-relevant passages, we explicitly model the posterior distribution $Q(z|x, y)$ with a learned neural model. Unlike the retriever $P_\eta(z|x)$, the label-posterior model has access to the target output and in hindsight can differentiate the label-relevant from other context-relevant passages. We learn the label-posterior jointly with the retriever and the generator by maximizing the evidence lower bound, ELBOLOSS, as given by the formula:

$$\log P(y|x) \geq \mathbb{E}_{z \sim Q(.|x,y)}[\log P_\theta(y|x, z)] - D_{\text{KL}}(Q\|P_\eta) \tag{2}$$

The ELBOLOSS has two terms with useful inductive biases. The first term maximizes the expectation of the generator's log-likelihood $P_\theta$ over the passages sampled from the label-posterior distribution $Q$. The generator need attend only to the label-relevant passages, biasing it toward relying more on the retrieved passages rather than its internal language model. The second term is the KL divergence from the retriever to the label-posterior, also referred to as the reverse KL divergence:

$$D_{\text{KL}}\big[Q(z|x, y) \mid P_\eta(z|x)\big] = \sum_{z \sim Q(.|x,y)} Q(z|x, y)\big(\log Q(z|x, y) - \log P_\eta(z|x)\big)$$

This term is again weighted by $Q(z|x, y)$, making it like a probabilistic implication: high $Q(z|x, y)$ implies high $P(z|x)$, i.e., label-relevance implies context-relevance but not vice-versa. In one-to-many tasks, which have many context-relevant passages but few label-relevant passages, this term captures the intuition that the retriever be penalized heavily if it doesn't retrieve the label-relevant passage but lightly if it retrieves other context-relevant passages that happen to not be label-relevant.

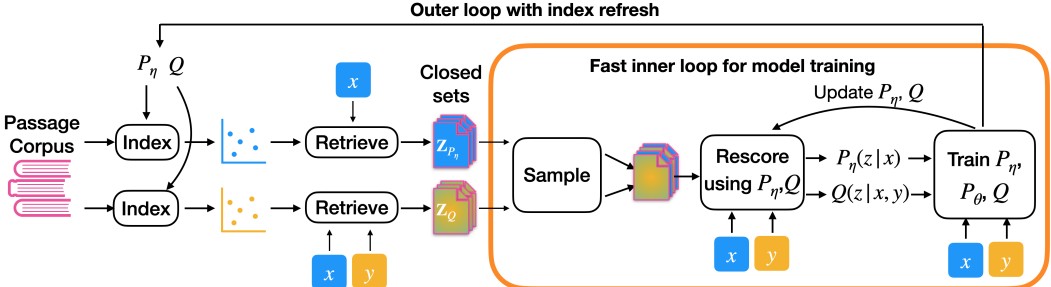

Figure 2: An overview of iterative closed-set training: We iterate through the outer-loop and call each execution a round. At the beginning of the round we re-index the passage corpus using the latest retriever $P_\eta(z|x)$ and guide-retriever $Q(z|x, y)$ to create a high-recall closed-set of top-$r$ passages for each retriever and query. Then, in the fast inner loop, we train the models for multiple epochs by sampling passages from the fixed closed-set and recomputing the probability distributions. The trained models are then used in the next round.

**Posterior as a retriever**     Rather than modeling the label-posterior $Q(z|x, y)$ as a re-reranker (that merely reranks documents as retrieved by the retriever $P_\eta$), we model it as a guide retriever that finds label-relevant passages from the entire corpus. We sample passages from the label-posterior distribution, and estimate the ELBoLoss more accurately than using passages from $P_\eta(z|x)$. The guide retriever generalizes weak supervision approaches (Lee et al., 2019; Guu et al., 2020) and relevance-guided supervision (Khattab et al., 2021), to **posterior-guided supervision** with a learned posterior retriever rather than brittle heuristics based on word-overlap.

**Iterative closed-set training**     Prior works (Guu et al., 2020; Khattab et al., 2021) intermittently update the passage index during training. To allow for such a workflow, we organize our training into rounds (see Figure 2). At the beginning of each round, in the outer loop, we encode the passages and the queries with various retrievers and find the highest scoring $r$ passages that we dub the closed-set. In the inner loop that runs for many epochs, we sample $k (= 8)$ passages from the closed-set ($r = 100$). This is fast because we are no longer retrieving from the entire corpus in the inner loop and also sufficient because the closed-set has a high recall. During the inner loop, we update the retrievers (both document and query encoders) and use the latest model parameters for computing the loss functions. A round results in trained models that are then used for the next round. We find that 2 rounds are often sufficient, with decreasing marginal utility from the third round onward.

**Distributional repositioning before inference**     We approximate the expectation terms in EL-BoLoss by sampling $k$ passages from the closed-set $Q_{\text{top-r}}(.|x, y)$, which provides better supervision than $P_\eta(z|x)$ and leads to faster training. However, the models only ever get exposed to passages from the $Q(.|x, y)$ distribution, which limits their ability to generalize over passages from $P_\eta(.|x)$ during inference. To remedy this, we instead sample passages from an $\alpha$-mixture of the two distributions: with probability $\alpha$, $z \sim P_\eta(.|x)$ and with probability $1 - \alpha$, $z \sim Q(.|x, y)$. In the initial rounds we set low values of $\alpha$ and increase it toward the end to reposition the passage distribution and better match with $P_\eta(.|x)$ at test time. The retriever and the generator can be trained by sampling passages from different $\alpha$-mixtures and we utilize this to avoid retriever overfitting (with $\alpha = 1$) while maintaining generator groundedness (with $\alpha = 0.25, 0.5$).

**Training individual models to convergence**     In practice, the retriever and the generator when jointly trained converge at different times. The single loss term in MARGINALIZEDLOSS (Eq. 1) hides the convergence of individual models; one model starts to overfit while the other model still hasn't converged. With ELBoLoss, there are two terms in Eq. 2: the first connecting $Q(z|x, y)$ and $P_\theta(y|x, z)$, the second connecting $Q(z|x, y)$ and $P_\eta(z|x)$. After training for a few epochs, we freeze the guide and train the models independently until convergence based on their individual losses.

## 4    EXPERIMENTAL EVALUATION

We evaluate on two open-ended knowledge-intensive tasks: informative conversations and free-form question answering. We ask the following three research questions:

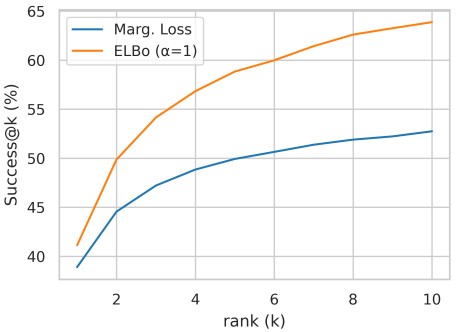 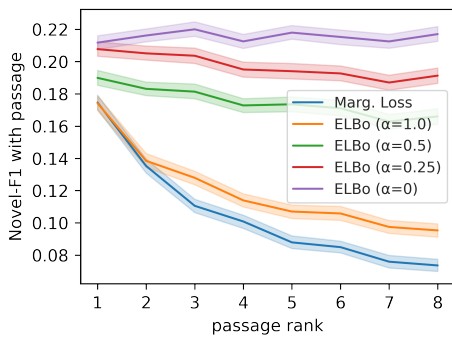

Figure 3: Relevance and Groundedness of models trained on the Wizard of Wikipedia dataset: (**left**) success@k of retrieved passages w.r.t. rank and (**right**) Novel-F1 between decoded output and retrieved passage w.r.t. retrieved passage rank. The ELBOLOSS retriever is more effective at retrieving the gold passage than the MARGINALIZEDLOSS retriever, especially when we consider the top-10 passages for this one-to-many task. The ELBOLOSS generators have higher overlap with top-k retrieved passages and the overlap increases as $\alpha$ decreases.

**RQ₁** Relevance: Are the retrieved passages more relevant? (Section 4.4)

**RQ₂** Groundedness: Does the generator make better use of the retrieved passages? (Section 4.5)

**RQ₃** Generation Quality: Does this lead to better end-to-end performance? (Section 4.6)

## 4.1 MODELS

**Retriever Models**     We model the retriever $P_\eta(z|x)$ and the guide-retriever $Q(z|x, y)$ using Col-BERT (Khattab & Zaharia, 2020). ColBERT encodes the query tokens $q_i$ and the document tokens $d_j$ independently using BERT, normalizes to produce unit-vectors $E_{q_i}$ and $E_{d_j}$, and defines similarity as $S_{q,d} = \sum_i \max_j E_{q_i}^T E_{d_j}$. Unlike DPR's [CLS] token embedding (Karpukhin et al., 2020), with ColBERT's late-interaction paradigm the query and document tokens retain their identities and contribute to a finer-grained term-wise similarity leading to state-of-the-art retrieval results on open-domain QA benchmarks (Khattab et al., 2021). To convert similarity scores into a probability distribution, calculate the softmax of the scores over the $k$ sampled passages. For the posterior-retriever, we concatenate the input and the output into the query $q = [x\, y]$. ColBERT pre-trained on the MS-MARCO passage ranking dataset is widely used for other tasks and we use it for the Wizard of Wikipedia task. However, the MS-MARCO NLGen task contains queries from the passage ranking pre-training dataset. Therefore, following Lewis et al. (2020), we use Natural Questions to pre-train ColBERT Khattab et al. (2021) for the MS-MARCO NLGen task.

**Generation Model**     Following Lewis et al. (2020) we use a pre-trained BART model and fine-tune it for the respective tasks during training. It is conditioned on both the context and the document and trained to produce the target. At test time, we decode using beam-search with a beam size of 4.

## 4.2 TASKS

**Informative conversations**     Informative conversations are open-ended because people have the agency to drive them in different directions at every turn (one-to-many) and are knowledge-intensive because the utterances contain specific bits of world knowledge. We evaluate with Wizard of Wikipedia (WoW) dataset (Dinan et al., 2019), where an "apprentice" chats (via text) with a "wizard", being curious about different topics, and the "wizard" grounds their response in a sentence from Wikipedia. The input for this task is the conversational history $x$, the output is the wizard's utterance $y$ and the models can retrieve individual passages ($z$) from all of Wikipedia ($\approx$26 million passages). We use the version of this dataset provided in the KILT benchmark (Petroni et al., 2021) and report leaderboard performance on the held out test set. We use the dev set to answer the granular research questions.

Table 1: Relevance evaluation: Our method (ELBoLoss Retriever, $\alpha = 1$) strongly improves over the baseline (MARGINALIZEDLOSS Retriever) for the one-to-many Wizard of Wikipedia dataset, in particular for $k = 5, 10$. The ELBo posterior finds $z_{gold}$ with high success providing better supervision during training. (MRR = Mean Reciprocal Rank, Success@k both in percentages.)

| Method | Wizard of Wikipedia | | | | MS MARCO NLGen | | | |
|---|---|---|---|---|---|---|---|---|
| | MRR | S@1 | S@5 | S@10 | MRR | S@1 | S@5 | S@10 |
| Gold-sup. | 45.2 | 35.6 | 57.0 | 63.1 | 28.9 | 19.5 | 40.4 | 47.7 |
| Marg. | 43.8 | 38.9 | 49.9 | 52.8 | 30.4 | 19.4 | 43.4 | 53.2 |
| ELBo ($\alpha = 1$) | **49.0** | **41.1** | **58.8** | **63.9** | **32.1** | **21.2** | **45.3** | **54.4** |

**Free-form Question Answering**     We use the MS-MARCO NLGen dataset (Nguyen et al., 2016) where the task is to generate natural-sounding answers to questions. This free-form open-QA task is one-to-one but more challenging than other *extractive* open-domain QA datasets. The dataset is a subset of MS-MARCO questions whose answers were reviewed by a separate editor and rewritten if they had a high overlap with one of the provided passages (indicating that the original editor may have copied the passage directly). These "well-formed answers" are meant to be complete sentences (such as can be read out by a conversational assistant) and are long (median length 11 words). The input for this task is a query $x$, the output is a well-formed answer $y$, and the models can retrieve from the MS-MARCO passage collection (8.8 million web passages). The public benchmark and the test set is no longer available for evaluation. Instead we split the public validation set into a validation and test set and show results on the test set.

While both datasets annotate the passages referred to by the person who wrote the target output (gold passages), we only use them for evaluation and validation and not for training.

### 4.3    BASELINES

Apart from the two main methods – MARGINALIZEDLOSS and ELBoLOSS – we train two additional baselines: **gold-supervised** and **generator-only**. For the Gold-supervised baseline, we assume that the gold-passage $z_{gold}$ is available during training and train a retriever by maximizing its log-likelihood. We take random passages from ELBo Retriever's closed-set as negatives (excluding top-10 to avoid false negatives). We train the Gold-supervised generator by simply maximizing $P_\theta(y|x, z_{gold})$. For the generator-only baseline, we ignore the existence of passages and directly maximize $P(y|x)$ with a sequence to sequence model.

### 4.4    RELEVANCE EVALUATION

We evaluate the relevance of the retrieved passages (**RQ$_1$**) using the gold passage labels supplied by each dataset. For one-to-many tasks, we expect the label-relevant passage to be one of the top-$k$ ($k = \{1, 5, 10\}$) retrieved passages. Thus, we report Success@$k$ (S@$k$ for short)[1], the percentage of inputs for which any gold provenance passage is retrieved within the top-$k$ ($k = \{1, 5, 10\}$) passages. We also report Mean Reciprocal Rank (MRR), a common IR evaluation metric.

Our results are shown in Table 1. With Wizard of Wikipedia, ELBoLOSS retriever markedly outperforms MARGINALIZEDLOSS. Both systems have a relatively high Success@1 and easily handle 38.9–41.1% of the examples, but the ELBoLOSS retriever continues to find many more relevant passages at larger retrieval depths $k$ widening the gap to 11 points for Success@10. With MS MARCO NLGen, ELBoLoss outperforms MarginalizedLoss by 1–2 points across our metrics, reflecting smaller—but nonetheless consistent—gains when compared with Wizard of Wikipedia, a one-to-many generation task. Based on manual inspection, we find many false negatives (corroborated by Arabzadeh et al. (2021)), i.e. passages that contain the answer but aren't marked as gold, leading to a lower Success@1 compared to Wizard of Wikipedia.

---

[1]With a single gold passage Recall@k and Success@k are numerically identical and sometimes used interchangeably; we prefer S@k because it is less ambiguous and widely used in the IR community since Voorhees (2004).

Table 2: Groundedness evaluation: Our method ELBoLoss ($\alpha = 0.25, 0.5$) shows more overlap between generated output and the retrieved passage than MARGINALIZEDLOSS and for the Wizard of Wikipedia dataset the gap increases as we consider the maximum over top-5 passages. (Novel-F1: discounts commonly occurring words and context words ($x$)).

| Dataset | Method | Top-1 | | Max. of Top-5 | |
|---|---|---|---|---|---|
| | | F1 | Nov-F1 | F1 | Nov-F1 |
| WoW | Gold-sup. Generator | 18.84 | 17.12 | 32.47 | 31.92 |
| | Marg. Generator | 18.63 | 17.46 | 26.19 | 25.39 |
| | ELBo Generator ($\alpha = 0.25$) | **21.34** | **20.78** | **34.16** | **34.24** |
| MSM | Gold-sup. Generator | 36.13 | 28.34 | 49.29 | 43.45 |
| | Marg. Generator | 33.12 | 25.39 | 45.76 | 39.45 |
| | ELBo Generator ($\alpha = 0.5$) | **34.52** | **26.49** | **46.91** | **40.47** |

**Effect of the guide-retriever**     MARGINALIZEDLOSS depends on the retriever $P_\eta(z|x)$ to find label-relevant passages during training and is therefore recall limited. MARGINALIZEDLOSS's success@100 on Wizard of Wikipedia saturates at 55.8% (not reported in the table) without much hope of further improvement because $z_{gold}$ that are never retrieved cannot provide positive examples for supervision. With ELBoLoss, the guide-retriever retrieves label-relevant passages with >85% success@10 (Table 8) for both the datasets providing better supervision than MARGINALIZEDLOSS. Consequently, ELBoLoss retriever's success@5 is higher than MARGINALIZEDLOSS despite containing $20\times$ less passages and reaches 69.3% for success@100.

**Comparison with Gold-supervised retriever**     We find that the Gold-sup. retriever quickly overfits during training resulting in a lower performance than using ELBo loss. We also see lower performance when training with ELBoLoss while sampling purely from $Q(.|x, y)$ (i.e., $\alpha = 0$, Table 7) because it is low-entropy and the same label-relevant passages get repeatedly sampled. By sampling from $P_\eta(.|x)$, the KL divergence is minimized over a wider and realistic support set of passages. Perhaps, there are many passages that have some label-relevant phrases making them partially relevant and $Q(z|x, y)$ "teaches" $P_\eta(z|x)$ to capture these phrase-level relative differences. Sampling from $P_\eta(.|x)$ leads to better generalization, has similarities to distillation and is an interesting direction for future work.

Overall, we find that ELBoLoss improves relevance of retrieved passages over MARGINALIZEDLOSS for two qualitatively different tasks, with larger gains for the one-to-many generation task.

## 4.5   GROUNDEDNESS EVALUATION

We now examine **RQ**$_2$, studying the degree to which the generator relies on the retrieved passages for producing its output. To quantify this *groundedness*, we compute F1-overlap between a *retrieved passage* (not necessarily the gold passage) and the produced text when generation is conditioned on that passage. We get the retrieved passages for each method (except generator-only) using the corresponding retriever.

As an analogue of Success@k, we propose *Max. F1@k*, the largest F1-overlap exhibited by any generated output with the corresponding retrieved passage fed to the generator. We also propose Novel-F1, a new metric that discounts words that occur frequently and words that already appear in the context $x$, since otherwise these tokens dominate raw F1 in practice (up to 80%, see Appendix A.5) but are not indicative of grounding.

Our results are shown in Table 2 and Figure 3. For the Wizard of Wikipedia dataset, we observe that our ELBoLoss generator outperforms MARGINALIZEDLOSS by 2.7 F1 (14.5% relative improvement) and 3.3 Novel-F1 (19% r.i.) when given the top retrieved passage. For the MS MARCO NLGen dataset, we observe smaller but consistent gains in groundedness (1–2 F1, Novel-F1) with ELBoLoss compared to MARGINALIZEDLOSS. In Figure 3 (right), MARGINALIZEDLOSS generator's overlap decays rapidly beyond the top passage, whereas the ELBoLoss ($\alpha = 0.25$) generator's overlap declines gradually. This shows that the ELBoLoss generator stays grounded beyond just

the top passage, a desirable property in one-to-many generation systems. We also see (in Figure 3, right) that groundedness increases as $\alpha$ decreases. However, a generator trained with $\alpha = 0$, despite being maximally grounded (Appendix table 5), has lower end-to-end performance (Appendix table 6) because it is unduly "trusting" of the provided passage (nearly flat line in Figure 3) and does not abstain from using irrelevant passages. We see the same effect with the Gold-supervised generator on MS-MARCO NLGen: it is more grounded but has lower downstream performance.

Overall, we find that ELBoLoss improves grounding of the generator over MARGINALIZEDLOSS for two qualitatively different tasks, with larger gains for the one-to-many generation task.

### 4.6 END-TO-END EVALUATION

To evaluate the end-to-end quality of our systems, we calculate F1 and Novel-F1 (defined in Section 4.5) of the decoded output with the *human-written gold output*. To allow for the possibility of the generator using any part of the *gold passage* (and not just the human-written gold output) for the Wizard Of Wikipedia task, we use Knowledge-F1 (defined by Shuster et al. (2021)): F1 between the sampled generation and the *gold passage*. Since it is reasonable to expect the gold passage to be in the top-$k$ for $k>1$ for one-to-many tasks (as in Section 4.5), we also compute the max. over top-$k$ retrieved passages.

The results are summarized in Table 3. For the Wizard of Wikipedia dataset, using only the top retrieved passage ELBoLoss leads to $6.7\%$ relative improvement in Novel-F1@1. But in the one-to-many setting, the label-relevant passage is an arbitrary choice from amongst the context-relevant passages. We account for that using max. overlap over the top-5 passages and see larger improvements for ELBoLoss , namely 1 F1, 2 Novel-F1 ($\sim 15\%$ r.i.), and 1.5 K-F1 ($\sim 10\%$ r.i.). For MS Marco NLGen, we see a small but consistent increase due to ELBoLoss over MARGINALIZED-LOSS: 1.5 F1 and 2 Novel-F1 across passage depths.

We also submit the above models (ELBoLoss and MARGINALIZEDLOSS) to the Wizard of Wikipedia task on the KILT leaderboard. ELBoLoss consistently outperforms the baseline MARGINALIZEDLOSS across all metrics (see Table 4). The table also reports Recall@5, which evaluates retrieval at a coarser granularity, namely at the *full Wikipedia page* level, though so far we have investigated it directly at the passage level. Consistent with the results in Table 1, our method also outperforms MARGINALIZEDLOSS in retrieval metrics. In fact, our ELBoLoss model achieves state-of-the-art performance across all the generation metrics (F1, ROUGE-L, KILT-F1, KILT-ROUGE-L) on the leaderboard, though it is not the strongest on R-Prec and Recall@5.[2]

To conclude, we have evaluated the ELBoLoss and MARGINALIZEDLOSS using a one-to-one free-form QA dataset and a one-to-many dataset of informative conversations. Our results show that our method ELBoLoss trains a better retriever, a more grounded generator and improves end-to-end performance, especially in the one-to-many setting.

## 5 DISCUSSION

**Hallucination, grounding and correctness**    Shuster et al. (2021) show that providing retrieved passages to a generator reduces hallucination. Our work increases grounding in the retrieved passage, promising to further reduce hallucination. Even though the generator is now more likely to use content from the provided passage (rather than hallucinating from parametric memory), that does not guarantee *correctness*. Our token-level overlap metrics that evaluate grounding do not capture this aspect either. There is scope for future work to address this gap with better training methods (and evaluation metrics) that produce (and reward) grounded and correct outputs.

**Practical matters: Trust and control**    For tasks like QA, we want a "conservative" generator: it should abstain from using a passage that doesn't contain the answer. For more open-ended tasks like informative conversations, we want the generator make use of diverse passages. In our work, we show that by reducing $\alpha$, the distribution of the passages shifts from $P_\eta(.|x)$ to $Q(.|x, y)$ and the generator increasingly trusts the retrieved passages (Figure 3). System designers can use the

---

[2]Earlier results on the KILT leaderboard for Wizard of Wikipedia should be interpreted with caution, as the KILT authors recently updated the train/dev splits due to anomalies in the preprocessing script. We have used the updated version for our model and baseline.

Table 3: End-to-end evaluation: Our method ELBoLoss improves over MARGINALIZEDLOSS when considering Max. overlap of generated output with target output over top-5 passages for the Wizard of Wikipedia dataset and also for top-1 with MS Marco NLGen dataset. (Novel-F1: discounts commonly occurring words and context words ($x$), Knowledge-F1: overlap of generated output with gold passage.)

| Dataset | Method | Top-1 | | | Max. of Top-5 | | |
|---------|--------|------|------|------|------|------|------|
| | | F1 | N-F1 | K-F1 | F1 | N-F1 | K-F1 |
| WoW | Gold-sup. | 16.70 | 8.53 | 11.64 | 24.95 | 14.87 | 16.16 |
| | Gen. Only | 16.11 | 5.15 | 8.05 | – | – | – |
| | Marg. | 18.79 | 10.45 | 12.61 | 26.52 | 16.42 | 16.02 |
| | ELBo | **18.86** | **11.12** | **13.08** | **27.56** | **18.67** | **17.69** |
| MSM | Gold-sup. | 59.25 | 36.22 | – | 71.44 | 55.02 | – |
| | Gen. Only | 51.75 | 14.71 | – | – | – | – |
| | Marg. | 60.18 | 37.19 | – | 72.22 | 56.06 | – |
| | ELBo | **61.46** | **39.65** | – | **73.18** | **58.19** | – |

Table 4: Wizard of Wikipedia KILT leaderboard evaluation: ELBoLoss achieves SoTA on generation metrics (F1, ROUGE-L, KILT-F1, KILT-ROUGE-L indicated with †) as of Oct 2021 and improves relevance over MARGINALIZEDLOSS

| | R-Prec | Recall@5 | F1 | ROUGE-L | KILT-F1 | KILT-ROUGE-L |
|---|--------|----------|------|---------|---------|--------------|
| Re2G (prev. best) | **60.10** | **79.98** | 18.90 | 16.76 | 12.98 | 11.39 |
| Marg. | 53.94 | 68.12 | 18.11 | 16.21 | 11.78 | 10.47 |
| ELBo (curr. best) | 56.08 | 74.26 | **19.19**[†] | **17.06**[†] | **13.39**[†] | **11.92**[†] |

$\alpha$-mixture as a tool to modulate the degree of trust placed by the generator in the retrieved passages. Further, when a "trusting" generator is deployed in real-life settings (e.g., in open-domain socialbots; Paranjape et al. 2020), external business logic can select an appropriate passage from the top-$k$ retrieved passages and effectively control the generated content.

**Comparison with Fusion-in-Decoder** Izacard & Grave (2021a) provide multiple passages to the generator simultaneously and in Izacard & Grave (2021b) they use the decoder's attention weights over each passage for relevance supervision. However, Sachan et al. (2021) use Fusion-in-Decoder only at inference, forgoing the decoder's attention weights and using an equivalent version of the MARGINALIZEDLOSS for training the retriever. In our experiments we found marginal improvements while using Fusion-in-decoder at inference and we believe our approach of conditioning on individual passages is more precise for relevance supervision in one-to-many generation tasks.

**Inductive biases of HINDSIGHT** A learned $P_\eta(z|x)$ that maximizes ELBoLoss (a lowerbound) will also maximize MARGINALIZEDLOSS. It can therefore seem counter-intuitive at first that EL-BoLoss empirically outperforms MARGINALIZEDLOSS. We hypothesize that it better captures the necessary inductive biases (refer Section 3). The $\alpha$-mixture further loosens the bound theoretically, inducing a bias toward the test distribution and improving generalization. Trading off theoretical tightness of bounds for desirable inductive biases proved fruitful for us and we hope that it inspires other researchers.

In this paper, we propose HINDSIGHT, a system that introduces a guide-retriever to improve supervision for both the retriever and the generator for retrieval-augmented, open-ended generation. During training, the guide retriever uses the target output of each example in order to find relevant passages, leading to better retrieval and more grounded generation. The resulting system achieves considerable empirical improvements over existing work, improving retrieval quality by up to 23%, grounding by up to 19%, and end-to-end output quality by up to 6.4%.

## ACKNOWLEDGEMENTS

We would like to thank the anonymous reviewers and Stanford colleagues: Eric Mitchell, Nandita Bhaskhar, Shikhar Murthy, Amelia Hardy and Antoine Bosselut for their valuable feedback. This research was supported in part by Stanford HAI, Samsung Electronics Co., Ltd, and affiliate members and other supporters of the Stanford DAWN project—Ant Financial, Facebook, Google, and VMware—as well as Cisco, Virtusa, SAP, and the NSF under CAREER grant CNS-1651570. Any opinions, findings, and conclusions or recommendations expressed in this material are those of the authors and do not necessarily reflect the views of the National Science Foundation.

## REPRODUCIBILITY STATEMENT

Section 3 contains key details of our method. The code for recreating these experiments along with hyperparameters will be released at `https://github.com/AshwinParanjape/hindsight`.

## ETHICS STATEMENT

Any generative system that produces free-form text and interacts with humans needs to have safeguards in place against harmful speech. In our work, we try to reduce hallucination and increase grounding which is a step in the right direction. It also helps restrict the content and exert control over the generation. But as we point out in the discussion (Section 5) increased grounding does not imply correctness. These systems by themselves are not ready to be deployed in the real world and future work needs to address outstanding issues.

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

# A APPENDIX

## A.1 IS HIGHER GROUNDING PURELY DUE TO A BETTER RETRIEVER?

To test this hypothesis, we ran an ablation test, where we used the best retriever we had (from Hindsight, $\alpha = 0$) and trained a generator using Marginalized loss while keeping the retriever itself fixed. This simulates the situation where we improved retrieval independently and we want to test if that is sufficient to explain increased grounding. We see in table 5 that that the better retriever in the ablation (Marg. Gen. with fixed ELBo Ret.) leads to increased grounding compared to the generator trained using MARGINALIZEDLOSS. However, the ablation is worse than ELBo Gen. ($\alpha = 0.25$), demonstrating that the increased grounding is also due to ELBoLOSS.

## A.2 WITHOUT DISTRIBUTIONAL REPOSITIONING

ELBoLOSS doesn't perform as well without distributional repositioning.

Table 5: Additional Groundedness evaluation for Wizard of Wikipedia: Models in descending order of groundedness. ELBoLOSS ($\alpha = 0.25$) has best tradeoff between groundedness and end-to-end evaluation.

| Method | Top-1 | | Max. of Top-5 | |
|---|---|---|---|---|
| | F1 | Nov-F1 | F1 | Nov-F1 |
| ELBo Gen. ($\alpha = 0$) | 21.48 | 21.18 | 35.04 | 35.27 |
| ELBo Gen. ($\alpha = 0.25$) | **21.34** | **20.78** | **34.16** | **34.24** |
| Marg. Gen. with fixed ELBo Ret. | 20.17 | 19.28 | 31.08 | 31.09 |
| Marg. Gen. | 18.63 | 17.46 | 26.19 | 25.39 |

Table 6: Additional End-to-end evaluation for Wizard of Wikipedia (Novel-F1: discounts commonly occurring words and context words ($x$), Knowledge-F1: overlap of generated output with gold passage)

| Method | Top-1 | | | Max. of Top-5 | | |
|---|---|---|---|---|---|---|
| | F1 | N-F1 | K-F1 | F1 | N-F1 | K-F1 |
| Marg. | 18.79 | 10.45 | 12.61 | 26.52 | 16.42 | 16.02 |
| ELBo ($\alpha_{\text{ret}} = 1, \alpha_{\text{gen}} = 0.25$) | **18.86** | **11.12** | **13.08** | **27.56** | **18.67** | **17.69** |
| ELBo ($\alpha_{\text{ret}} = 1, \alpha_{\text{gen}} = 0$) | 18.41 | 11.03 | 12.93 | 27.04 | 18.13 | 17.61 |
| Gen. Only | 16.11 | 5.15 | 8.05 | – | – | – |

Table 7: Additional Relevance evaluation: ELBoLOSS Retriever with $\alpha = 1$ is better than $\alpha = 0$. Sampling passages with temperature $= 4$ helps with overfitting but still performs worse than $\alpha = 1$. (MRR = Mean Reciprocal Rank, Success@k both in percentages)

| Method | Wizard of Wikipedia | | | |
|---|---|---|---|---|
| | MRR | S@1 | S@5 | S@10 |
| Marg. Retriever | 43.8 | 38.9 | 49.9 | 52.8 |
| ELBo Retriever ($\alpha = 1$) | **49.0** | **41.1** | **58.8** | **63.9** |
| ELBo Retriever ($\alpha = 0$, temperature=4) | 43.7 | 35.4 | 53.5 | 60.4 |
| Gold-sup. Retriever | 45.2 | 35.6 | 57.0 | 63.1 |
| ELBo Posterior | 78.5 | 72.4 | 86.0 | 88.4 |

Table 8: The ELBo posterior finds $z_{gold}$ with high success providing better supervision during training. (MRR = Mean Reciprocal Rank, Success@k both in percentages)

| Method | Dist. | Wizard of Wikipedia | | | | MS MARCO NLGen | | | |
|--------|-------|-----|-----|-----|------|-----|-----|-----|------|
| | | MRR | S@1 | S@5 | S@10 | MRR | S@1 | S@5 | S@10 |
| Gold-sup. | $P_\eta(z\|x)$ | 45.2 | 35.6 | 57.0 | 63.1 | 28.9 | 19.5 | 40.4 | 47.7 |
| Marg. | $P_\eta(z\|x)$ | 43.8 | 38.9 | 49.9 | 52.8 | 30.4 | 19.4 | 43.4 | 53.2 |
| ELBo ($\alpha = 1$) | $P_\eta(z\|x)$ | **49.0** | **41.1** | **58.8** | **63.9** | **32.1** | **21.2** | **45.3** | **54.4** |
| ELBo ($\alpha = 1$) | $Q(z\|x,y)$ | 78.5 | 72.4 | 86.0 | 88.4 | 67.8 | 56.7 | 81.9 | 86.2 |

Table 9: Passages retrieved by ELBoLoss retriever while talking about Italian Cuisine. They include passages about various ingredients (rank=2), cheeses (rank=4), dishes (rank=5) alongside more information about Bucatini Pasta (rank=1,3).

| rank | text |
|------|------|
| 1.0 | Bucatini > Abstract ǀ Bucatini , also known as perciatelli , is a thick spaghetti-like pasta with a hole running through the center. The name comes from , meaning "hole", while "bucato" or its Nea... |
| 2.0 | Pasta con le sarde > Ingredients. ǀ The principal ingredients are olive oil, onions, pasta and a finely chopped mixture of sardines and anchovy. Various types of pasta are used for the dish, but b... |
| 3.0 | Bucatini > Preparation. ǀ Standard pasta machines will roll out sheets of flat pasta which are then cut into ribbons to make flat, ribbon-style pasta like fettuccine, tagliatelle, or pappardelle. ... |
| 4.0 | Bocconcini > Abstract ǀ This cheese is described by its Italian name, which means "small mouthfuls". It is made in the "pasta filata" manner by dipping curds into hot whey, and kneading, pulling, ... |
| 5.0 | Carbonara > Abstract ǀ Carbonara () is an Italian pasta dish from Rome made with egg, hard cheese, guanciale (or pancetta), and black pepper. The dish arrived at its modern form, with its current ... |

## A.3 EXAMPLES OF RETRIEVED PASSAGES

### A.3.1 CONVERSATION 1: ITALIAN CUISINE

**Other** Ooh I like that! Stick some nice spicy arrabbiata sauce with it, ahhhh! Have you ever had bucatini before?

**Self** Oh yeah! I love that spicy garlic and tomato sauce. No I have not had bucatini. Is that a type of cheese?

**Other** Now you're speakin' my language. No no, it's a style of noodle, like a really long straw. Bucatini amatraciana is insanely good.

Table 10: Passages retrieved by MARGINALIZEDLOSS retriever while talking about Italian Cuisine. All passages talk about Pastas.

| rank | text |
| --- | --- |
| 1.0 | Bucatini > Abstract \| Bucatini , also known as perciatelli , is a thick spaghetti-like pasta with a hole running through the center. The name comes from , meaning "hole", while "bucato" or its Nea... |
| 2.0 | Bucatini > Preparation. \| Standard pasta machines will roll out sheets of flat pasta which are then cut into ribbons to make flat, ribbon-style pasta like fettuccine, tagliatelle, or pappardelle. ... |
| 3.0 | Rotini > Abstract \| Rotini is a type of helix- or corkscrew-shaped pasta. The name comes from a 17th-century Italian word meaning "small wheels". Rotini is related to fusilli, but has a tighter he... |
| 4.0 | Vermicelli > History.:The Americas. \| The "fideo" is a type of noodle, produced in Europe ever since the Roman times, best known as "fideus" or "fidelis", and then spread to Mexican and Latin Amer... |
| 5.0 | Rollatini > Abstract \| Rollatini (sometimes also spelled rolatini or rolletini) is an Italian-style dish (called "rollatini di melanzane" in faux Italian) that is usually made with thin slices of ... |

### A.3.2 CONVERSATION 2: ROCK AND ROLL

**Self**  Do you mean Elvis Aaron Presley, the American singer and actor?

**Other**  That's the one. I think his nickname was the king of rock 'n roll.

**Self**  I had just heard of him being "The King". There probably would not have been a Sun Records if not for Elvis and Sam Phillips.

**Other**  He was revolutionary for his time. Many older people thought he was straight from the devil.

Table 11: Passages retrieved by ELBoLOSS while talking about Rock and Roll. Relevant passages about cultural impact of Elvis Presley (rank=2) and details about his career (rank=4) alongside introductary paragraphs of other musicians

| rank | text |
|---|---|
| 1.0 | Sam Phillips > Abstract \| Samuel Cornelius Phillips (January 5, 1923 – July 30, 2003) was an American record producer who played an important role in the development of rock and roll during the 19... |
| 2.0 | Cultural impact of Elvis Presley > Abstract \| Since the beginning of his career, Elvis Presley has had an extensive cultural impact. According to "Rolling Stone", "it was Elvis who made rock 'n' r... |
| 3.0 | Freddie King > Abstract \| Freddie King (September 3, 1934 – December 28, 1976) was an American blues guitarist and singer. He recorded several hits for Federal Records in the early 1960s. His soul... |
| 4.0 | Elvis Presley > Abstract \| With a series of successful network television appearances and chart-topping records, he became the leading figure of the newly popular sound of rock and roll. His energ... |
| 5.0 | Elvis Presley > Abstract \| Elvis Aaron Presley (January 8, 1935 – August 16, 1977), also known mononymously as Elvis, was an American singer, musician, and actor. Regarded as one of the most signi... |

Table 12: Passages retrieved by MARGINALIZEDLOSS while talking about Rock and Roll. All passages are the introductory paragraphs from various related artists

| rank | text |
|---|---|
| 1.0 | Elvis Presley > Abstract \| Elvis Aaron Presley (January 8, 1935 – August 16, 1977), also known mononymously as Elvis, was an American singer, musician, and actor. Regarded as one of the most signi... |
| 2.0 | Sam Phillips > Abstract \| Samuel Cornelius Phillips (January 5, 1923 – July 30, 2003) was an American record producer who played an important role in the development of rock and roll during the 19... |
| 3.0 | Johnny Otis > Abstract \| Johnny Otis (born Ioannis Alexandres Veliotes; December 28, 1921 – January 17, 2012) was an American singer, musician, composer, arranger, bandleader, talent scout, disc j... |
| 4.0 | Carl Perkins > Abstract \| Called "the King of Rockabilly", he was inducted into the Rock and Roll Hall of Fame, the Rockabilly Hall of Fame, the Memphis Music Hall of Fame, and the Nashville Songw... |
| 5.0 | Chubby Checker > Abstract \| Chubby Checker (born Ernest Evans; October 3, 1941) is an American rock 'n roll singer and dancer. He is widely known for popularising many dance styles including the t... |

### A.4 EXAMPLES OF GENERATED OUTPUTS

We look at a conversation that illustrates the differences in grounding between MARGINALIZED-LOSS generator and ELBoLOSS generator. We see that the MARGINALIZEDLOSS generator hallucinates in 4 out of 5 utterances and only uses retrieved passage content in 2 out of 5 utterances. On the other hand ELBoLOSS generator doesn't hallucinate and uses retrieved passage content for all utterances.

Prior turns:

**Self** I'm still working on getting better proper nutrition (Cheetos get me) but moderate vigorous exercise feels great

**Other** Yes, we have a swimming pool so I've been able to get a bit more exercise, while enjoying the cool water during this hot summer.

Table 13: Utterances generated by MARGINALIZEDLOSS generator based on top 5 retrieved passages (passage text omitted for brevity). Phrases in the utterance that are based on the retrieved passage are marked in bold. Hallucinated phrases are italicised.

| passage title | text |
|---|---|
| Physical fitness > Abstract | That's great! I love swimming too. *It's a great way to stay hydrated* and **prevent hypokinetic diseases**. |
| Outline of Exercise > Abstract | That's great! I love swimming too. It's a great way to keep your **immune system healthy and prevent diseases like heart disease and diabetes**. |
| Underwater hockey > Abstract | That's great! I love swimming too. *It's a great way to keep your body hydrated.* |
| Exercise Trends > Abstract | That's great! I love swimming too. *It's a great way to keep your body hydrated.* I've never been to a pool before. |
| Callanetics > Abstract | That's great! I love swimming pools too. They're great for exercise, especially in the summer. |

Table 14: Utterances generated by ELBOLOSS generator based on top 5 retrieved passages (passage text omitted for brevity). Phrases in the utterance that are based on the retrieved passage are marked in bold. Hallucinated phrases are italicised.

| passage title | text |
|---|---|
| Physical fitness > Abstract | That's great! I'm also dedicated to **physical fitness**, which is a **state of health and well-being. It's important to be healthy, to resist hypokinetic diseases**. |
| Exercise > Abstract | That's awesome! I've been doing it for a long time. It's a great way to **improve strength, prevent aging, develop muscles and the cardiovascular system**. |
| Bodybuilding > Abstract | That's awesome! I'm also into **bodybuilding**, which **is the use of progressive resistance exercise to control and develop one's musculature for aesthetic purposes**. |
| General fitness training > Abstract | That's great! I'm also into **general fitness training**, which is focused on **broad goals of overall health and well-being**. |
| Strength training > Abstract | That's awesome! I'm also into **strength training**, which is a type of **physical exercise specializing in the use of resistance to induce muscular contraction**. |

## A.5 NOVEL-F1

**Rationale** We conducted a small experiment with the generated output on Wizard of Wikipedia dataset using top-8 retrieved passages. We removed the gold passage and computed overlap of the generated output with the target output. We consistently found (across models and passage ranks) the F1 overlap to be close to 15. This meant that by conditioning on arbitrary passages the generator (likely by ignoring them altogether) is able to achieve around 80% of the F1-overlap of the best performing models ($\sim 19$ F1). This can be a confounding factor for selecting models based on high F1 overlap. A model that simply copies content from the input $x$ can achieve high F1-overlap but fail to using the retrieved passage to generate the output. Removing commonly occurring words

reduces it to 8 F1, but removing words from input context reduces it further down to 4 F1. Thus we find Novel-F1 to be the cleanest measure of overlap as it discounts two confounding factors and only looks at "Novel" words, words that are rare and were not in the input text $x$.

We construct the list of common words based on their frequency in the training corpus. We sort words by frequency and take the most frequent words that contribute: 50% of the probability mass toward Wizard of Wikipedia utterances (amounting to 121 words) following Shuster et al. (2021). However, we found that using the same heuristic for MS-MARCO NLGen answers included numbers and rarer tokens that could potentially be in the answer span. So we instead use only 33% of the probability mass (amounting to 55 words). We also ran evaluation using 50% of the probability mass but found the trends to be consistent.

**MS Marco NLGen list of common words**    (sorted by frequency)
is, of, in, to, and, for, or, are, that, on, from, as, by, you, with, it, county, can, at, per, was, your, average, cost, be, between, which, used, one, united, states, there, years, located, name, not, new, have, takes, number, has, means, days, when, blood, system, year, should, no, most, first, hours, up, minutes, 1

**Wizard of Wikipedia list of common words**    (sorted by frequency)
is, of, in, to, and, for, or, are, that, on, from, as, by, you, with, it, county, can, at, per, was, your, average, cost, be, between, which, used, one, united, states, there, years, located, name, not, new, have, takes, number, has, means, days, when, blood, system, year, should, no, most, first, hours, up, minutes, 1 i, and, of, in, is, to, it, that, are, you, they, have, was, but, for, as, its, like, with, on, so, be, or, not, yes, do, can, from, there, by, well, also, one, my, know, has, some, he, their, love, most, people, think, really, all, about, just, too, them, im, which, sure, more, been, at, would, many, were, good, very, dont, when, thats, no, yeah, what, other, great, if, because, used, actually, first, since, lot, me, even, your, how, we, time, different, world, use, get, called, only, out, much, over, had, though, music, around, popular, his, am, made, than, such, back, up, us, make, usually, who, favorite, new, food, oh, long, she, now, did, pretty, any, where, years, this, way, go

## B  INTUITION BEHIND IMPROVEMENTS DUE TO ELBoLoss

To understand the intuition behind suboptimality of MARGINALIZEDLOSS for open-ended generation tasks consider the following: We would want a good retriever to assign similar but high probabilities to all context-relevant passages because they are similarly relevant but a good generator to only assign high probabilities when using label-relevant passages because only label-relevant passages are pertinent to the target output. But the training signal to a model (partial derivative w.r.t the model and a passage) is modulated by the probability of the other model:

$$\frac{\partial P(y|x)}{\partial P_\eta(z_i|x)} = P_\theta(y|x, z_i) \qquad\qquad \frac{\partial P(y|x)}{\partial P_\theta(y|x, z_i)} = P_\eta(z_i|x)$$

Since context-relevant passages have similar $P(z_i|x)$ the gradient encourages the generator to assign equal probabilities to the target output using all context-relevant passages. We see this issue play out empirically when using MARGINALIZEDLOSS for two different tasks: Open-Domain QA (Natural Questions by Kwiatkowski et al. (2019)) and informative dialogue (Wizard of Wikipedia by Dinan et al. (2019)) (Figure 4). We see that on the Natural Questions dataset, where there is typically one correct answer, the generator produces distribution with a sharp peak that can potentially serve as an accurate proxy for label-relevance and in turn train a good retriever. But on the Wizard of Wikipedia dataset, the generator produces a flatter distribution which is a bad proxy for label-relevance. This provides weaker supervision for the retriever which learns a flatter probability distribution as well and is less able to differentiate context-relevant from irrelevant passages.

We see in Figure 5 that for the Wizard of Wikipedia dataset with ELBoLoss we obtain a sharp distribution for $Q(z|x, y)$ (nearly as good as $P_\theta(y|x, z)$ on NQ from Figure 4) and that the $P_\eta(z|x)$ and $P_\theta(y|x, z)$ are now sharper than MARGINALIZEDLOSS. While a sharper distribution does not imply a better retriever and generator (they may still assign high probability to the wrong passage), a flatter distribution limits their potential. As we will see in Section 4, ELBoLoss indeed utilizes the potential and trains a better retriever and more a grounded generator.

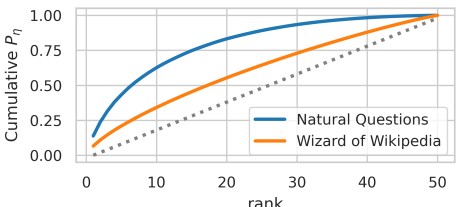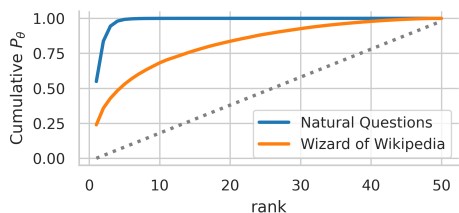

Figure 4: With MARGINALIZEDLOSS, the generator $P_\theta(y|x, z)$ learns a sharp distribution for Natural Questions (NQ) dataset (**right**) but learns a flatter distribution for a one-to-many open-ended generation task using the Wizard of Wikipedia dataset (WoW). The flatter distribution in the case of WoW Generator shows that it has not learned label-relevance as well. Consequently, for WoW we see a weaker retriever (**left**) that has a flatter distribution than NQ. (**Left**) Cumulative probability $P_\eta(z|x)$ w.r.t. rank for passages. (**Right**) Assuming a uniform prior $P(z|x)$, the cumulative probability $P_\theta(y|x, z)$ w.r.t. rank for passages, plotted as $P(z|x, y) \propto P(y|x, z)P(z|x)$. The gray dotted line shows a hypothetical model that assigns equal probabilities to all passages.

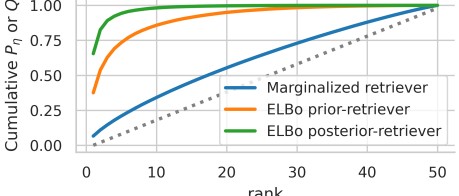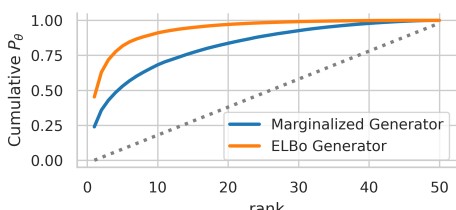

Figure 5: Analogous plots to Figure 4 but with ELBOLOSS on the one-to-many Wizard of Wikipedia (WoW) dataset. Training with ELBOLOSS produces a sharp distribution for $Q(z|x, y)$ and subsequently sharper $P_\eta(z|x)$ and $P_\theta(y|x, z)$ than MARGINALIZEDLOSS.

