# OpenReview forum: "Hindsight: Posterior-guided training of retrievers for improved open-ended generation"
_ICLR.cc/2022/Conference — ICLR 2022 Poster_

### Official Review · Reviewer_uUxg · 2021-10-25

**Correctness:** 3
**Technical Novelty And Significance:** 3
**Empirical Novelty And Significance:** 2
**Recommendation:** 6
**Confidence:** 3

**Main Review:**

The strength of this paper is:
They novelly proposed to train a dedicated module, "guide-retriever", to predict the knowledge passage probability with extra information (target) and use that to guide (via KL divergence) the retriever. It seems to be a reasonable approach.
The evaluation and experiments are also relatively comprehensive.

The weakness is that this paper only compares their method with one baseline (MARGINALIZEDLOSS), but there're more baselines are designed for the similar tasks and should be analyzed: e.g., "REALM: Retrieval-Augmented Language Model Pre-Training" by Guu et al., and "Retrieval-augmented generation for knowledge-intensive nlp tasks" by Lewis et al.

**Summary Of The Paper:**

This work focuses on the knowledge grounded text generation tasks. They argue that multiple passages can be valid and relevant to the context, but not all of them are observed/used in the target response. Therefore they propose that, during training, the target response should be utilized to train a "guide-retriever",  which predicts P(passage|context, response) and provides passage weight for the generator. A retriever is jointly trained with the generator and the guide-retriever, and the KL divergence between the retriever and the guide-retriever is included in the loss function.

The contribution of this paper is that they proposed a potential solution to the challenge that the model may be not effectively trained in case that multiple passages are valid but only one of few are used.

**Summary Of The Review:**

This work proposes a novel approach to the knowledge grounded text generation tasks via the design of a new "guide-retriever".
The methodology is reasonable and the experiments show significant improvement over one baseline. However, this work can be improved if comparison with more baselines is provided.

---

> ### Author Response · Authors · 2021-11-20
> **Addressing a lack of baselines**
>
> Thank you for the positive feedback and suggestions.
>
> The main concern raised was regarding a lack of baselines.
>
> 1 - We would like to point out that Marginalized Loss is essentially the same training objective used in "Retrieval-augmented generation for knowledge-intensive nlp tasks" by Lewis et al.  Therefore we think that we have compared with that work but we will make it more explicit in the final version.
>
> 2 - "REALM: Retrieval-Augmented Language Model Pre-Training'' provides a method to pre-train retrievers. However, it does not offer any method for generation. In our work we address open-ended generation tasks and we therefore don’t think that ReaLM is an appropriate baseline for our work.
>
> 3 - During the review period, we have added a baseline/ablation where gold-passages supervise the models in [the common reply to all reviewers](https://openreview.net/forum?id=Vr_BTpw3wz&noteId=_2b2aA-cPCV) . We also now have a no-retriever (only-generator) baseline ([post link](https://openreview.net/forum?id=Vr_BTpw3wz&noteId=MW4Pbe5q33W)) and an ablation where the generator is trained using the (fixed) hindsight retriever using Marginalized loss ([post link](https://openreview.net/forum?id=Vr_BTpw3wz&noteId=xGV9e74i87J)).
>
> We hope that the additional baselines increase your confidence in our results.

---

### Official Review · Reviewer_eXap · 2021-10-27

**Correctness:** 4
**Technical Novelty And Significance:** 3
**Empirical Novelty And Significance:** 2
**Recommendation:** 8
**Confidence:** 4

**Main Review:**

# Main Review

### Strengths

- The authors indeed find improvement over a strong baseline (using a marginalized loss instead) in all three points across two unrelated datasets.
- The proposed models are straightforward and not overly complex, with a proper diagram easily describing the training procedure, thus making the methods reproducible.
- The findings are relevant and important particularly for those interested in what the authors describe as “one-to-many” generation tasks, where several plausible outputs can stem from a single input; the proposed model helps to narrow the “many” possible generations to the one most similar to the gold human response.
- The submission is quite clear; methods (and the model architecture) are proposed clearly and concisely, providing the reader with appropriate background and technical knowledge to understand the model. The results are plainly displayed in tables in figures and are clearly described in the text.

### Weaknesses
- I think the authors failed to consider the model on an important subset of tasks that, in my opinion, would benefit most from this procedure: tasks without **gold retrieved passages available**. Though the authors compare to a “MarginalizedLoss” baseline, as it is used in some strong models in the literature, the real ablation to test is retrievers trained **_directly with the gold passage as target_**. WoW and MS-MARCO both provide such passages, and yet in the paper they are only used for evaluation of trained models. A more interesting task would be to apply these models to datasets without such provided targets.
- Utilizing the target output to improve knowledge-grounded dialogue is not particularly novel in the literature:
    - [1] proposes a system (TPPA) that learns to retrieve knowledge both from the input context and the response
    - [2] utilizes the responses to inform the model where to pay attention to in the given knowledge, and thus can obtain models that ground more appropriately in the knowledge
- The authors claim that a main contribution of the method is a model that produces generations more grounded in the retrieved passages than other methods. However, one could argue this naturally follows from the simply improving retrieval in the first place, in that the model will ground more often if it receives less loss from doing so, and it will receive less loss if it uses more relevant passages. So the overarching contribution here is improving retrieval through posterior-guided supervision, the effectiveness of which could be better supported if other situations were considered, see above..

# Questions + Feedback
### Questions
1. Did you experiment with not updating the index, and instead just updating query encoders as in [3]?
2. Why call your metric “success@10”, rather than “recall@10”? Are these fundamentally different metrics?
3. Why were different pre-trained ColBERT models used for each task?
4. In section 5, paragraph “Comparison to Fusion-in-Decoder”, you mention that “Fusion-in-Decoder is uniquely useful for QA style tasks”, however it is in fact used directly on the Wizard of Wikipedia task in [4] and achieves the best results reported there (better, e.g., than marginalized-loss methods such as RAG).

### Minor Feedback
1. Typo end of section 2: “we describe our method we separates those two concerns”
2. Section 3, “Posterior as Retriever Paragraph”: “...and estimate the ELBoLoss ‘more’ accurately…”
3. End of section 4.2: “... only use them for evaluation and validation and not for training nor for validation”, this sentence is unclear
4. It would be nice to introduce “r.i.” as an abbreviation of “relative improvement” before it is introduced in section 4.4


# References
[1] Zheng et. al. 2020, Approximation of Response Knowledge Retrieval in Knowledge-grounded Dialogue Generation

[2] Zheng et. al. 2021, Knowledge-Grounded Dialogue Generation with Term-level De-noising

[3] Lewis et. al. 2021, Retrieval-Augmented Generation for Knowledge-Intensive NLP Tasks

[4] Shuster et. al. 2021, Retrieval Augmentation Reduces Hallucination in Conversation


**Summary Of The Paper:**

The paper tackles the problem of open-ended knowledge-grounded natural language generation, in the context of free-form QA or knowledge-grounded dialogue, where models must ground their generations on passages relevant to the input context. Specifically, the authors explore improving the retrieval component of retrieval-augmented systems by utilizing posterior signal from the label. A “guide retriever” learns the relevant passage to retrieve by including the target output in its input context, and then, using an ELBo loss, provides this signal to the normal, in-system retriever. The authors find that their model improves three-fold over baselines: 1) it retrieves relevant passages more frequently; 2) its generations are more grounded in retrieved passages; and 3) its generations are closer to human generations.


**Summary Of The Review:**

The authors show improvements in two downstream tasks when improving retrieval via posterior-guided supervision of a retriever in a retrieval-augmented generation setup. However, the datasets considered are not explored to their maximum potential, with appropriate baselines missing from the paper (training the retrievers directly on the target passages); and, datasets where this solution appears to be most applicable are not considered at all (those without target passages). The paper indeed supports its claims, as mentioned in the summary, though some work is missing that could bolster the proposed efficacy of the method.

---

> ### Author Response · Authors · 2021-11-20
> **Addressing the source of increased groundedness and other questions**
>
> > The authors claim that a main contribution of the method is a model that produces generations more grounded in the retrieved passages than other methods. However, one could argue this naturally follows from the simply improving retrieval in the first place, in that the model will ground more often if it receives less loss from doing so, and it will receive less loss if it uses more relevant passages.
>
> To test this hypothesis, we ran an ablation test, where we used the best retriever we had (from Hindsight, $\alpha=0$) and we trained a generator using Marginalized loss while keeping the retriever itself fixed. This simulates the situation where we improved retrieval first and we want to test if the generator learns to ground more often. We compare it with the groundedness evaluation from Table 2 for Wizard of Wikipedia and find that the ablation is less grounded.
>
> |  | Top-1 F1 |Top-1 Novel-F1 | Top-5 F1 | Top-5 Novel-F1 |
> |----|-----|------|------|-------|
> | Marg. Loss | 18.63 | 17.46 | 26.19 | 25.39 |
> | ELBo Loss | 21.34 | 20.78 | 34.16 | 34.24 |
> | Marg. Loss with fixed best retriever | 20.17 | 19.28 | 31.08 | 31.09 |
>
> This is also corroborated by Figure 3 (right) where we see that lower $\alpha$ (i.e.. sampling documents from the posterior distribution more frequently) increases groundedness even when using ELBo Loss. Thus, while we improve retrieval through posterior-guided supervision, we also independently increase groundedness. We think this is an important clarification, and we thank the reviewer for pointing it out. We will include this ablation in the paper.
>
> > Did you experiment with not updating the index, and instead just updating query encoders?
>
> We agree that it is an important scientific question: whether fixing the index hurts as it will reduce the time required to re-index the corpus. However, we did not experiment with keeping the index fixed as it is a weaker version of our model in principle and is orthogonal to the overall contribution. ColBERT uses a single shared encoder for encoding both the query and the document. As a result, keeping the document encoder fixed while updating the query encoder would require a non-trivial amount of engineering effort. We are working on releasing this code publicly and will include the possibility of fixing the index as it is also computationally efficient.
>
> > Why call your metric “success@10”, rather than “recall@10”? Are these fundamentally different metrics?
>
> This concern is shared by another reviewer as well and we have responded to it in full in a common response above. In short, what we want to measure is whether we got at least one gold passage in the top-k, even if there were multiple gold annotated passages. This is consistent with the one-to-many formulation of the task. This is best described as success@k, and is better suited than recall@k which may be understood as the average of retrieved over gold per example. (Incidentally, in our datasets, nearly all queries have a single gold passage, and thus success and average recall would be numerically quite similar.)
>
> > Why were different pre-trained ColBERT models used for each task?
>
> This concern was shared by other reviewers and we have addressed it in a common response.
>
> > In section 5, paragraph “Comparison to Fusion-in-Decoder”, you mention that “Fusion-in-Decoder is uniquely useful for QA style tasks”, however it is in fact used directly on the Wizard of Wikipedia task and achieves the best results reported there (better, e.g., than marginalized-loss methods such as RAG).
>
> We agree with the reviewer that Fusion-in-Decoder does indeed achieve the best results and is therefore a better generation model as measured by the automatic metrics on the Wizard of Wikipedia dataset. However, when deployed to realistic systems, such as open-domain chatbots, it is desirable to condition on retrieved passages separately, generate responses and then select the final response using external mechanisms (e.g. known-user interests, topicality, avoiding profanity, etc). In our own experiments using FiD, we found the model to largely be acting as a reranker and focusing on a single retrieved passage. This makes the generations less diverse in terms of content, which is essential to deployed systems.
>
> We meant for this discussion point to be from a real-world application perspective and not about model performance. We would like to thank the reviewer for raising this question and we would address it by writing it clearly.

---

> > ### Comment · Reviewer_eXap · 2021-11-23
> > **Update Following Author Response**
> >
> > Thank you for your comprehensive response; this is a reply to this comment as well as the one above (and the general responses to all reviewers)
> >
> > TL;DR: I am satisfied with the majority of your responses and have raised my score. More details below.
> >
> > 1. _Concerning retrievers trained with gold passage as target_ -- your experiments have addressed this issue.
> > 2. _Concerning evaluation on datasets without gold passages_ -- your point that evaluations such as Knowledge-F1 and retriever performance in general require gold passages is fair. However, while downstream end-to-end evaluation is indeed noisy, this is ultimately an evaluation that we are very, if not the most, interested in -- and I think you would agree, given that the title of the paper includes "... improved open-ended generation" :). So, it is still an important research question, but perhaps is appropriate future work.
> > 3. _Concerning novelty_ -- thank you for addressing your place in the related literature.
> > 4. _Concerning increased groundedness_ -- I appreciate these additional experiments, as they do appear to support your claims.
> >
> > Additionally, I feel that you have appropriately addressed my other questions.

---

> ### Author Response · Authors · 2021-11-20
> **Addressing lack of tasks without gold passages and the novelty of our method**
>
> Thank you for the thorough review of our work. We are glad that you found our work relevant and important, our results/baselines strong, and our presentation clear.
>
> > Though the authors compare to a “MarginalizedLoss” baseline, as it is used in some strong models in the literature, the real ablation to test is retrievers trained directly with the gold passage as target. WoW and MS-MARCO both provide such passages, and yet in the paper they are only used for evaluation of trained models.
>
> We have now conducted this comparison and found the retrievers trained directly with the gold passage as the target are noticeably less effective. We explain this in more detail in a common reply to all reviewers above.
>
> > I think the authors failed to consider the model on an important subset of tasks that, in my opinion, would benefit most from this procedure: tasks without gold retrieved passages available. A more interesting task would be to apply these models to datasets without such provided targets.
>
> Gold retrieved passages are essential for evaluation purposes, as we would not be able to compute retriever performance or Knowledge-F1 otherwise. However, by not using the gold passages during training, we are simulating the scenario where the gold passages are not available. Of course, our method is most useful in practice when gold passages are truly unknown—but for evaluation with no gold passages, we would have to rely purely on end-to-end evaluation, which is known to be noisy for open-ended generation tasks. We are indeed looking into future work where the gold passages are unknown and we welcome any suggestions in this space.
>
> >Utilizing the target output to improve knowledge-grounded dialogue is not particularly novel in the literature:
> [1] proposes a system (TPPA) that learns to retrieve knowledge both from the input context and the response
> [2] utilizes the responses to inform the model where to pay attention to in the given knowledge, and thus can obtain models that ground more appropriately in the knowledge
>
> Thank you for bringing these papers to our attention. There has indeed been prior work on utilizing target output and we would like to add them to the background section.
>
> However, in TPPA, the task definition is different and Kp is provided to the model. Kp is a small set of passages (\~50) containing the gold passage and therefore Recall@50 is 100%. In our setting, we do not assume such a set Kp, but rather retrieve from the entire Wikipedia (\~22M) passages or MS-MARCO dataset (\~7M). Simply using the retriever to sample passages (i.e. Marg. Loss) is insufficient e.g. Recall@100 plateaus at \~55% for the Wizard of Wikipedia dataset. On the other hand, even if we were to use BM25 (from TPPA) to find the most relevant passage from the entire retrieval corpus it is quite likely to lead to false positives. This has been observed in open-QA training where many passages containing the answer string are irrelevant to the question. [2] looks at grounding within a single passage, whereas we are trying to improve grounding when considering multiple passages.
>
> While the general idea of using the target output is not in itself novel, our work has theoretically motivated (ELBo) novel ideas about training a posterior retriever that is able to retrieve from the entire corpus while also learning a scoring function (compared to BM25 that is static), and also using passages from the guide/posterior retriever to train a generator that is more grounded.
>
> [1] Zheng et. al. 2020, Approximation of Response Knowledge Retrieval in Knowledge-grounded Dialogue Generation
>
> [2] Zheng et. al. 2021, Knowledge-Grounded Dialogue Generation with Term-level De-noising

---

### Official Review · Reviewer_hgu8 · 2021-11-02

**Correctness:** 3
**Technical Novelty And Significance:** 2
**Empirical Novelty And Significance:** 2
**Recommendation:** 6
**Confidence:** 4

**Main Review:**

--------------
Strengths
--------------
1. Improvement in performance over a baseline RAG model by use of the posterior for training the retriever
2. Experiments on aspects related to retrieval as well as response generation
--------------
Weakness
---------------
1. Limited evaluation - no experiments to study hallucination/memorization. This is a big weakness because BART is a powerful generation model.

----------------
Questions:
----------------
1. The paper discusses using an alpha-mixture but doesn't clearly explain the need for the same. I'd imagine there's a risk of not seeing the ground-truth document when sampling from P_eta (because of a poorly trained prior network) and that is probably why alpha is set low initially, but what happens if you only sample from Q? Perhaps experiments that demonstrate the benefit of using an alpha-mixture over just using the posterior Q could be useful.
2. What happens if the ground-truth document is never sampled; the KL divergence term will not add anything meaningful in that case? Is there a risk of posterior collapse? Do you do anything in your training to address this problem? This is also tied to my comment about hallucination; in the event that the model does not see the "correct" document while training the retrievers, it will force the generator to learn without grounding.
3. There is contemporaneous unpublished work submitted with perhaps the exact same approach (with minor differences in how the approximation is done):  https://openreview.net/forum?id=vdTCHL1nyEq Were the authors aware of this work?
4. Why was pre-training of the retrievers done differently?

**Summary Of The Paper:**

In this paper authors describe an approach to use the responses/ answers to guide retrieval during training of document grounded response generator. By having the retriever being trained using the posterior (p(document|response,context)), the model could learn a better (supervised) retrieval network (based on ColBERT) which could be used to guide the training of the prior ( p(document|context)).  Documents are retrieved using the dialog context and the top-k documents are used for training the posterior as well as the prior. Since the expectation for ELBOLoss cannot be computed exactly (due to the large document set),  it is computed by sampling documents from the top-r retrieved documents -- from either the posterior or the prior, guided by a parameter called alpha which governs the sampling proportion. The idea of doing variational training using a posterior network isn't particularly novel but the authors have made it work for open-ended response generation using this approximation for computing the ELBOLoss. The networks use BART for language generation and CoLBERT for modeling the retrievers. Experiments have been presented using the Wizard of Wikipedia dataset and the MSMARCO NLGEN. Experiments show an improvement over the baseline model (RAG - referred to as MarginalizedLoss) in both retrieval (success@k, MRR) as well as response generation (text-F1 overlap between response and grounded document and textF1 overlap between response and ground-truth output response). Overall this is a well written easy to read paper



**Summary Of The Review:**

This is a simple and interesting idea which shows promise in performance. However, the experimental results are a little in weak -- in particular there is no study about hallucination/memorization in these models and some of the modeling choices aren't well studied.

---

> ### Author Response · Authors · 2021-11-20
> **Addressing hallucination/memorization due to BART and what happens when we purely sample from Q**
>
> We thank you for the valuable feedback. We are glad that you found our idea interesting and liked the evaluation across retrieval as well as response generation.
>
> > Limited evaluation - no experiments to study hallucination/memorization. This is a big weakness because BART is a powerful generation model.
>
> It is indeed a plausible hypothesis that because BART is a powerful pre-trained generation model, it might be able to remember world knowledge from pre-training and therefore generate correct output based on world knowledge from parametric memory or hallucinate in its absence.
>
> BART is a powerful generator, but it is not sufficiently powerful for “knowledge-intensive” generation tasks. To investigate this, we ran a baseline (“No retrieval baseline”) where we do not retrieve any passages and the generator is simply trying to maximize $P(y|x)$. We compare with the other models on the end-to-end metrics for the Wizard of Wikipedia dataset here (from Table 4). We plan to add this baseline to the paper.
>
> |    | F1 |  N-F1 |  K-F1 |
> |-----|-----|------|-------------------|
> |Marg. Loss | 18.79 | 10.45 | 12.61|
> |ELBo Loss  | 18.86 |11.12 |13.08 |
> |No retrieval baseline | 16.11 | 5.15 | 8.05 |
>
>
> We see that, in addition to a drop in F1, there is a steep drop in Novel-F1 and Knowledge-F1 for the baseline compared to the retrieval augmented generation methods. An increase in Knowledge-F1 is correlated with a decrease in hallucination [1] and therefore we would expect hallucination to reduce for ELBo Loss compared to Marg. Loss. Increased groundedness (overlap between generated utterance and retrieved passage) is also indicative of reduced hallucination (our paper, Table 2). While we have only used automatic metrics and do not have explicit evaluation for hallucination, the automatic metrics point toward reduced hallucination. If you believe additional results are necessary to address this question, we can run a small-scale human evaluation to measure hallucination more directly.
>
>
> > but what happens if you only sample from Q? Perhaps experiments that demonstrate the benefit of using an alpha-mixture over just using the posterior Q could be useful
>
>
> We thank you for this thoughtful question. When we sample only from Q, the trained retriever and generator don’t generalize as well. At training time, the models are exposed to samples from label-posterior Q, but during inference the retriever finds top-k passages from the entire corpus.
>
> For the generator, we see that training over samples purely from Q makes it unduly “trusting” of the provided passages because it only saw label-relevant passages during training. Therefore at inference time, it does not abstain from using irrelevant passages, resulting in lower end-to-end performance.
>
> Table comparing end-to-end performance using the same (Hindsight) retriever but generators with different $\alpha$. Sampling purely from $Q$ corresponds to $\alpha=0$.
>
> | Model | F1| N-F1| K-F1 |
> |---------|----|------|-------|
> |ELBo Generator ($\alpha$=0.25) | 18.86 | 11.12 | 13.08 |
> | ELBo Generator ($\alpha$=0) | 18.41 | 11.03 | 12.93 |
>
> Sampling purely from Q is also detrimental for retriever training. We find that the label-posterior trains fast and once trained, it is sharp/peaky (low entropy, Appendix Figure 5 left). Sampling from low entropy distribution implies that the same passages get sampled repeatedly over multiple epochs. Thus, it is easy for the retriever to overfit those passages on the training examples. We tried increasing the temperature prior to sampling; this mitigates the effect to some extent. However, setting alpha to 1 for the last round of training was still more effective than sampling from Q.
>
> Table comparing two retrievers trained using ELBo objective with varying $\alpha$. $\alpha=0$ corresponds to sampling purely from $Q$.
>
> | Model | MRR | Success@1| Success@5 | Success@10|
> | -------|---------|----------|-------|---------|
> |ELBo Retriever ($\alpha$=1) | 49.0   |41.1 | 58.8  |63.9 |
> |ELBo Retriever ($\alpha$=0, passage_sampling_temperature=4) | 43.7 | 35.4 | 53.5 | 60.4 |
>
> We plan to add these tables to the appendix with a short discussion in the main paper.
>
> [1] Shuster, Kurt, Spencer Poff, Moya Chen, Douwe Kiela and Jason Weston. “Retrieval Augmentation Reduces Hallucination in Conversation.” EMNLP (2021).

---

> > ### Comment · Reviewer_hgu8 · 2021-11-23
> > **Updated Review Score**
> >
> > Thank you for your responses -- I have updated my review score.

---

> ### Author Response · Authors · 2021-11-20
> **Addressing posterior collapse and other questions**
>
> > What happens if the ground-truth document is never sampled; the KL divergence term will not add anything meaningful in that case?
>
> Even if the ground-truth document is never sampled, we find that in a realistic corpus such as Wikipedia, many non-gold passages are false negatives: they contain partial label-relevant information but aren’t labeled explicitly. The KL divergence is still meaningful in those cases.
>
> In fact, in the above experiment where we sample only purely from Q, the retriever overfits the training set very easily. However, with an appropriate $\alpha$-mixture the samples are diverse enough to prevent overfitting. One of the strengths of this method is that the KL divergence term effectively distills the posterior to the prior, as defined over many varied passages. Thus, we believe the KL divergence term is meaningful even when the ground-truth document is never sampled. This is also corroborated by a baseline retriever that we supervised with gold passages, which we present in a common reply to all reviewers.
>
> > Is there a risk of posterior collapse? Do you do anything in your training to address this problem?
>
> For a task where passages from the corpus don’t add value for a majority of the instances, there is a risk of posterior collapse that we do not address. A possible mitigation that future work might consider is to use a dummy passage, which doesn’t contain any text and denotes the case when no passage in the corpus is label-relevant. The posterior would learn to assign high probability to the dummy passage in such cases and avoid posterior from collapsing for the instances which have label-relevant passages.
>
> > This is also tied to my comment about hallucination; in the event that the model does not see the "correct" document while training the retrievers, it will force the generator to learn without grounding.
>
> Thank you for raising this important point. We will clarify this better in the paper. The retriever and generator can be trained using different alpha mixtures. In our case, the retriever was trained with $\alpha$=1 and the generator was trained with $\alpha$=0.25. Thus, the generator gets to see the “correct” document more often and learns to ground. At the same time, the retriever avoids overfitting and learns to generalize.
>
> > There is contemporaneous unpublished work submitted with perhaps the exact same approach (with minor differences in how the approximation is done):  https://openreview.net/forum?id=vdTCHL1nyEq Were the authors aware of this work?
>
>
> Thank you for bringing this to our attention. We were unaware of this contemporaneous work from the OpenReview Anonymous preprint server. The idea is indeed similar but our work demonstrates much stronger results, especially for retriever relevance. Our work uses different datasets and a late-interaction retriever model (ColBERT) with iterative index updates as opposed to DPR. As pointed out in the previous answer, a key salient contribution is distributional repositioning ($\alpha$-mixture) that is not used in the preprint. While it might seem like a minor difference in how the approximation is done, it is crucial to make the approach work well in practice. We also perform a granular evaluation, especially w.r.t to groundedness. Despite the similarities to this concurrent work, we believe that our contributions add significant value for the community.
>
> > Why was pre-training of the retrievers done differently?
>
> This concern was shared by other reviewers and we have addressed it in a common response.

---

### Official Review · Reviewer_kJHS · 2021-11-03

**Correctness:** 4
**Technical Novelty And Significance:** 4
**Empirical Novelty And Significance:** Not applicable
**Recommendation:** 8
**Confidence:** 4

**Main Review:**

Strengths:
* The problem in this paper is important and underexplored --- despite recent advances in neural retrieval, knowledge-grounded open-ended generation is less studied compared to other problems like short answer generation. As the paper motivates, there are many reasons that prior work focusing on short answer generation are not suitable for open-ended generation.
* The proposed method is novel, and shows very strong empirical results.

Weaknesses:
* One major concern is that the pipeline training baseline ---  the approach that trains the retriever and the generator separately --- is missing, although it is (arguably) the most widely-used approach and is a potentially stronger baseline. Pipeline training (Karpukhin et al, Khattab et al and more) has been much more widely used in recent retrieval work compared to joint training due to its simplicity, stability, and better empirical results (e.g., as shown in [1] which compares joint training and pipeline training under the same condition). Although I agree with the motivation for joint training in the paper, there is no justification for not adding a pipeline training baseline, especially given that the datasets come with annotated gold passage so it is pretty straightforward to train two models separately.
* Three research questions for evaluation in Section 4 all look reasonable to me, but how the specific metric used in the paper measures those is not entirely clear to me. For example, is F1 / Nov-F1 between retrieved evidence and generated response a standard metric to measure the groundedness? If not, there should be some evaluation that shows that F1 / Nov-F1 correlates with human evaluation of the groundedness of the generated response.

Minor comments / questions
* Is “Success @ k” an existing term taken from prior work? This metric is identical to “Recall @ k” which is a standard, long-standing term. Unless “Success” is another widely-used term as an alias of “Recall” that I am unaware of, it should be replaced to “Recall”.
* Is there a reason that ColBERT is pretrained on MS-MARCO or NQ? Has there been any ablation on the impact of this pretraining? Does any of findings in the paper change if the model is not pretrained on another dataset?

[1] https://arxiv.org/pdf/2101.00408.pdf

**Summary Of The Paper:**

The paper tackles a problem of knowledge-grounded open-ended generation and proposes a new model that is an extension of an end-to-end training of the retrieval and the generator. In particular, there is another model called “posterior-guide” that is jointly trained with the other two models using ELBo. Intuitively, this posterior-guide is similar to the retriever that scores the evidence, but conditioned on not only the question but also the response. It is used as a weight of each evidence in the generation in order to encourage the generator to ground more to the evidence that is more relevant to the response. The retriever is also trained to be close to the posterior-guide (minimizing KL-divergence). To my understanding, this is a very clever way to give supervision to the model when it is not easy to obtain distant supervision data (unlike in short answer generation where the gold evidence is easily obtained by whether the short answer is included in the evidence or not).

Experiments are done on Wizard of Wikipedia and MS Marco NLGen. The model achieves significant improvements over the baseline retriever, based on three evaluation metrics: relevance, groundedness and generation quality.


**Summary Of The Review:**

Overall a strong paper, tackling important and underexplored problem of knowledge-grounded open-ended generation, proposing novel objective, significant empirical improvements on two datasets in multiple metrics. There are a few concerns like absence of a potentially stronger baseline, justification of evaluation metrics, choice of terms, choice of implementation details.

---

> ### Author Response · Authors · 2021-11-22
> **Addressing pipeline training and other concerns**
>
> We thank you for providing a thorough review of our paper and the encouraging comments. We are glad that you found our paper well motivated and the results strong.
>
> We would like to address your main concerns below:
>
> > Although I agree with the motivation for joint training in the paper, there is no justification for not adding a pipeline training baseline, especially given that the datasets come with annotated gold passage so it is pretty straightforward to train two models separately.
>
> This was an oversight and we have now fixed it. The results for the Gold-supervised retriever are explained in a common response to all reviewers. Despite having access to gold-passages, surprisingly, pipeline training performs worse. We think this could be because it is easy for the retriever and the generator to overfit the gold-passage during training. With ELBo Loss we are sampling many passages during training the retriever and the generator do not overfit the training set as quickly and are better able to generalize. The fact that the passages are weighted by the label-posterior perhaps improves signal-to-noise ratio. This is a very interesting observation and something that needs to be more thoroughly investigated in future work. We would like to thank you for your suggestion to run the gold-supervised baseline and leading us to it.
>
> > is F1 / Nov-F1 between retrieved evidence and generated response a standard metric to measure the groundedness?
>
> Prior work has used F1 with the gold passage, a.k.a knowledge-F1 instead of F1 with the target [1]. However, to the best of our knowledge, no prior work has considered overlap between the retrieved evidence and generated response. We think that comparing the retrieved evidence and generated response is important for evaluating generators for open-ended generation tasks.
>
> > If not, there should be some evaluation that shows that F1 / Nov-F1 correlates with human evaluation of the groundedness of the generated response.
>
> We agree and would like to run a small human study testing the correlation between these automatic metrics and groundedness (pehaps in conjunction with human judgement of hallucination as suggested by another reviewer). It was not possible within the time constraint of the discussion period but we would like to add it soon. Based on manual inspection of the retrieved passages and the generated outputs, we found that Nov-F1 picks up unique phrases and is highly indicative of grounding. We will add some examples of generated outputs and their overlap with the retrieved passages to the appendix.
>
> > Is “Success @ k” an existing term taken from prior work? This metric is identical to “Recall @ k” which is a standard, long-standing term. Unless “Success” is another widely-used term as an alias of “Recall” that I am unaware of, it should be replaced to “Recall”.
>
> We have responded to this in a common response. “Success” is indeed a widely-used term in IR literature. In our case, where we want to measure if any of the gold passages are retrieved and not necessarily the proportion of gold passages retrieved in total. With a single gold passage, these two quantities are numerically identical, but "Success" is less ambiguous.
>
> > Is there a reason that ColBERT is pretrained on MS-MARCO or NQ? Has there been any ablation on the impact of this pretraining? Does any of findings in the paper change if the model is not pretrained on another dataset?
>
> Pre-training a retriever is a separate and computationally intensive process and was not the focus of this work. This is in line with prior work on retrieval-augmented generation [2] that also used a pre-trained retriever.
>
> Both MS-MARCO and NQ have more training instances (700k, 300k respectively) which means that the pre-trained models generalize better.  Had we not used pre-trained models then the quality of the closed-set passages in the first round is likely to be lower and we would expect that it takes many rounds to reach convergence and even then it might overfit and not reach the same level of performance.
>
> While it was outside the scope of this work, we are looking into future work that uses hindsight for pre-training over a large open-ended language dataset.
>
> [1] Shuster, Kurt, Spencer Poff, Moya Chen, Douwe Kiela and Jason Weston. “Retrieval Augmentation Reduces Hallucination in Conversation.” EMNLP (2021).
>
> [2] Lewis, Patrick, Ethan Perez, Aleksandara Piktus, Fabio Petroni, Vladimir Karpukhin, Naman Goyal, Heinrich Kuttler, Mike Lewis, Wen-tau Yih, Tim Rocktäschel, Sebastian Riedel and Douwe Kiela. “Retrieval-Augmented Generation for Knowledge-Intensive NLP Tasks.” ArXiv abs/2005.11401 (2020): n. pag.

---

### Author Response · Authors · 2021-11-20
**Addressing common reviewer concerns**

We would like to take a moment to thank the reviewers for spending their time and energy reviewing our paper. We found their comments and pointed questions to be pertinent to our paper and will make it significantly stronger. The concerns were laid out clearly and are some of the most constructive comments that we have received as authors.

The reviewers found knowledge-grounded open-ended generation to be a well motivated problem and the solution to be clever when direct retriever supervision isn’t possible. They found our proposed models to be straightforward and not overly complex, and acknowledged that we found improvement over a strong baseline in all three points across two unrelated datasets. Overall they found our paper to be well written and easy to read.

There were some shared concerns among many reviewers that we address below -

### 1 - *Gold supervised retriever*
(Addressing Reviewer kJHS & Reviewer eXap)

They noted that even though the task shouldn’t be using gold passages during training, we should still compare with the case where gold passages are used during training to supervise the retriever.

As an ablation, we trained a retriever using the gold passage as the positive example and random passages from top-100 passages from the closed-sets as negative examples (we excluded the top-10 to account for possible false negatives) by maximizing the log-likelihood of the positive passage.

Retriever performance on Wizard of Wikipedia dataset

|Method | MRR | S@1 | S@5 | S@10 |
|-|-|-|-|-|
|Marg. Retriever | 43.8 | 38.9 | 49.9 | 52.8 |
| ELBo Retriever | 49.0 |  41.1 |  58.8 |   63.9 |
| Gold-sup. Retriever | 45.2 | 35.6 | 57.0 | 63.1 |

Retriever performance on MS-MARCO NLGen dataset

|Method | MRR | S@1 | S@5 | S@10 |
|-|-|-|-|-|
|Marg. Retriever | 30.4|19.4|43.4|53.2 |
| ELBo Retriever | 32.1 |  21.2 |   45.3 |   54.4 |
| Gold-sup. Retriever | 28.9 | 19.5 | 40.4 | 47.7 |

A caveat is that we only ran the gold-supervised baseline for the final round using the closed-set as retrieved by the Hindsight method. This closed-set has a higher recall, thus it is possible that the numbers are higher than what they would be, had we run multiple rounds using only the gold-supervised baseline alone.

We see that the performance of gold-supervised baseline is much lower than Hindsight. We see that the retriever, during training, is easily able to memorize the gold-passage and therefore overfits. However, with Hindsight, the label-posterior is able to assign finer-grained posterior probabilities over many passages. We hypothesize that minimizing KL divergence with the retriever over a wider support set provides a richer training signal, generalizes better and ultimately prevents overfitting. We thank the reviewers for leading us to this interesting result which shows that the posterior-guided training might be helpful in more ways than originally proposed, but more investigation is necessary in future work.

### 2- Success@k v/s Recall@k
(Addressing Reviewer kJHS & Reviewer eXap)

For relevance, we would like to measure if a relevant passage was found in top-k. For the “one-to-many” setting, even if there are multiple relevant passages, we care about finding any one. Therefore success@k is more accurate and reflective of what we want to measure. We realize that in the NLP community recall@k is often used to mean success@k, however, success@k is used extensively in Information Retrieval literature [1,2] and is part of the official trec evaluation suite [3,4]. We feel that success@k is less ambiguous and we will add a clarifying footnote to the paper. If the reviewers feel that recall@k is better for clarity, we can make that change as well.

### 3 - Why were different pre-trained retrievers used for the two tasks?
(Addressing Reviewer hgu8 & Reviewer eXap)

The most commonly used version of pretrained ColBERT is trained using the MS-MARCO passage retrieval task. However, the MS-MARCO NLGen task uses a subset of the queries. During pre-training the ColBERT model would have seen the gold passages which would amount to cheating. Thus, for MS-MARCO NLGen, we use a separate ColBERT model that has been pretrained on the Natural Questions dataset instead. It is common practice in literature [5] to use a retriever pre-trained using Natural Questions dataset.

[1] Voorhees, Ellen M.. “Overview of the TREC 2004 Robust Retrieval Track.” TREC (2004).

[2] Khattab, O., Christopher Potts and Matei A. Zaharia. “Relevance-guided Supervision for OpenQA with ColBERT.” Transactions of the Association for Computational Linguistics 9 (2021): 929-944.

[3] https://github.com/usnistgov/trec_eval/blob/master/m_success.c

[4] https://pyterrier.readthedocs.io/en/latest/experiments.html#pyterrier.measures.Success

[5] Lewis, Patrick SH, Ethan Perez, Aleksandra Piktus, Fabio Petroni, Vladimir Karpukhin, Naman Goyal, Heinrich Küttler et al. "Retrieval-Augmented Generation for Knowledge-Intensive NLP Tasks." (2020).

---

### Author Response · Authors · 2021-11-22
**Summary of updates to the paper**

- Added missing prior work - Background section last paragraph
- Clarified that retriever and generator can be trained using different $\alpha$s - last line of "Distributional repositioning before inference"
- Clarified the reason for using NQ pretrained ColBERT for MS-MARCO NLGen
- Added Gold-supervised retriever results in Table 1 with discussion in paragraph "Comparison with Gold-supervised retriever"
- Added Gold-supervised generator groundedness results in Table 2, and gold-supervised pipeline system in Table 3
- Clarified reason for using success@k over recall@k
- Discussed $\alpha=0$ regime in the main text and added tables 5, 6, 7 to the appendix with numbers
- Added Generator-only baseline to Table 3
- Deleted confusing discussion about FiD and QA
- Added a section to Appendix on "Is higher grounding purely due to a better retriever?" with additional results in Table 5
- Added a section to Appendix with "Examples of Generated outputs" containing generations from Marg. Loss in Table 12 and ELBo Loss in Table 13.
- Typo fixes

---

### Decision · Program_Chairs · 2022-01-20

**Decision:**

Accept (Poster)

**Comment:**

The authors study the problem of open-ended knowledge-grounded natural language generation, in the context of free-form QA or knowledge-grounded dialogue, focusing on improving the retrieval component of the retrieval-augmented system. By retrieving more relevant passages, the generations are more grounded in retrieved passages.

Pros:
+ The paper is clearly written and motivated.
+ Presents a straightforward approach that shows improvement over a  strong baseline.
+ A strong paper focuses on a rather under-explored problem of knowledge-grounded open-ended generation, proposing novel objective, significant empirical improvements on two datasets in multiple metrics.
+ The authors included human evolution results to support their findings.
+ The authors did a good job addressing several questions raised during review period and added several new experiment results and discussions to strengthen their findings. The reviewer team was generally satisfied.

Cons:

+ Several related work on knowledge guided dialog response generation is missing in the paper. Although the paper's focus is on retrieval based QA systems, the main focus is on open domain generation, which has overlaps with dialog response generation research. So the authors should cite the following papers in their paper: [1] Dinan, Emily, et al. "Wizard of wikipedia: Knowledge-powered conversational agents." arXiv preprint arXiv:1811.01241 (2018). [2] Zhou, Kangyan, Shrimai Prabhumoye, and Alan W. Black. "A dataset for document grounded conversations." arXiv preprint arXiv:1809.07358 (2018). [3]Zhan, Haolan, et al. "CoLV: A Collaborative Latent Variable Model for Knowledge-Grounded Dialogue Generation." Proceedings of the 2021 Conference on Empirical Methods in Natural Language Processing. 2021. [4]Zhao, Xueliang, et al. "Knowledge-grounded dialogue generation with pre-trained language models." arXiv preprint arXiv:2010.08824 (2020).
+ There are several related work concerning with generation of a response given a relatively small set of evidence text such as the following ones:  [5]Lian, Rongzhong, et al. "Learning to select knowledge for response generation in dialog systems." arXiv preprint arXiv:1902.04911 (2019). [6]Kim, Byeongchang, Jaewoo Ahn, and Gunhee Kim. "Sequential latent knowledge selection for knowledge-grounded dialogue." arXiv preprint arXiv:2002.07510 (2020). Although these work do not include a retrieval part, the authors should cite and discuss similarities and differences to [5] & [6] in their paper.